



# Earth System Sensitivity: a Feedback perspective

Peter O. Passenier

Independent Human Factors Professional

*Correspondence to*: Peter O. Passenier (passenr@ziggo.nl)

**Abstract.** In the field of climate-change research a lot of effort is devoted to the 'narrowing down' of uncertainties in the estimation of Equilibrium climate sensitivity (ECS), the global mean warming as a result of an instantaneous doubling of the $CO_2$ concentration in the atmosphere. The present study explores possible consequences of this narrowing down of ECS for the long-term Earth system sensitivity (ESS), taking into account 'slow' feedbacks due to the cryosphere response (permafrost

melting and ice-sheet disintegration) to a warming world. Implications for international policy making, aiming at avoiding 2 degrees Celsius of global warming, are briefly discussed.

## 1 Introduction

A significant amount of effort in the field of climate-change research is devoted to the 'narrowing down' of uncertainties in the estimation of Equilibrium climate sensitivity (ECS), the global mean warming as a result of an instantaneous doubling of

the $CO_2$ concentration in the atmosphere (see, for instance, Brown et al., 2017 or Caldwell, 2018). The latest estimate of ECS by the Intergovernmental Panel on Climate Change (IPCC) ranges from 1.5–4.5 K (with a central value of 3 K) and has remained unchanged for more than 25 years (IPCC AR5, 2013).

A more recent attempt of narrowing down this range can be found in Cox et al. (2018), who constrain CMIP5 climate models by their ability to simulate observed variations in climate. This resulted in a central estimate of 2.8 K for 2×$CO_2$ and a 'likely

range' of 2.2–3.4 K. This value sits towards the middle to lower end of the current IPCC range, thus possibly increasing the feasibility of avoiding 2 degrees Celsius of global warming as required by the Paris agreement.

The present study aims to explore possible consequences of this narrowing down of ECS for the long-term Earth system sensitivity (ESS), taking into account 'slow' feedbacks due to the cryosphere response (permafrost melting and ice-sheet

disintegration) to a warming world. The rationale being the signs that such 'slow' feedbacks possibly already start to kick in (see, for instance, NASA: Global Climate Change vital-signs web portal for most recent facts and figures), potentially leading to a substantial increase of Earth's temperature. Hence, the fact that these effects are not accounted for by the present generation of climate models (such as the ones used in the recent Cox et al. 2018 study), but stand at the basis of international policy making, may be regarded as peculiar, if not worrisome.



An important aspect in this respect is the uncertainty about the timescale of the cryosphere response (how *slow* is 'slow'?). In line with the more 'traditional' point of view, the cryosphere response is considered to become relevant on a timescale of millennia (see Hansen, 2005, for an editorial essay on this topic). Following his reasoning however, due to the nonlinear characteristics of this process, we could very well talk about a response time of centuries or less. This puts the rather strict, but maybe artificial, separation between ESS and ECS as a metric for climate-change policy making in a different perspective. Because of this multiscale aspect of Climate vs Earth system sensitivity to greenhouse gas (GHG) forcing, a qualitative assessment of 'long-term' cryosphere interactions may be of interest.

The (second-order) cryosphere effects do not need to be explicitly modeled to show that something may be learned from the structural relationship between the different feedbacks involved. This is in line with the basic principles of feedback analysis, originating in Electrical Engineering and control systems as described by Roe in his pedagogic review of 2009. Using this feedback-analysis approach, principal scaling relations between variations in ECS and variations in ESS may be determined, focusing on a better mechanistic understanding of interactions *in* (as opposed to improving the predictability *of*) the Earth climate system. As such, the research reported here can be regarded as an application of the fundamental paper of Roe et al. (2007) on the (un)predictability of Climate sensitivity, to the domain of Earth system sensitivity.

## 2 Method

Climate sensitivity (S) is defined as the equilibrium global mean surface temperature change ($\Delta T_{eq}$) in response to a specified unit radiative forcing (F) according to (Hansen et al., 2012):

$$S = \Delta T_{eq}/F \tag{1}$$

This quantity S depends on climate feedbacks, making a distinction between the 'fast-feedback' Charney sensitivity (Charney, 1979) related to fast hydrological responses (water vapor, cloud and sea ice) and the 'long-term equilibrium' sensitivity, related to slow surface-albedo feedbacks (governed by changes of ice-sheet area and vegetation cover). These two components map onto ECS, the 'Equilibrium climate sensitivity', and ESS, the 'Earth system sensitivity'. In the present study the effect of melting permafrost (release of non-$CO_2$ trace gases such as methane) is treated as an additional feedback, thus resulting in the Earth system sensitivity to $CO_2$ forcing (in Hansen et al. 2012 referred to as '$S_{CO2}$').

To further elaborate on this climate-feedback perspective, the equilibrium global mean surface temperature change ($\Delta T_{eq}$) is written as (in accordance with Hansen et al., 2008):

$$\Delta T_{eq} = f \, \Delta T_0 \tag{2a}$$
$$= \Delta T_0 + \Delta T_{feedbacks}$$
$$= \Delta T_0 + \Delta T_1 + \Delta T_2 + \ldots,$$



where $\Delta T_0$ is the global mean surface temperature change in the absence of climate feedbacks (radiative blackbody damping only), f is the net feedback factor and the $\Delta T_i$ are increments due to specific feedbacks. As an alternative to the feedback factor f, Hansen introduced the gain g to clarify the role of climate-feedback processes:

$$g = \Delta T_{feedbacks}/\Delta T_{eq} \qquad (2b)$$
$$= (\Delta T_1 + \Delta T_2 + \ldots)/\Delta T_{eq}$$
$$= g_1 + g_2 + \ldots$$

$g_i$ is positive for an amplifying feedback and negative for a feedback that diminishes the response. The additive nature of the $g_i$, unlike $f_i$, is a very useful characteristic of the gain. Evidently

$$f = 1/(1 - g) \quad \text{or, conversely, } g = 1 - 1/f \qquad (3)$$

Note that some studies use a different (or even reversed) definition of gain and feedback factor (see, for instance, the review of Roe (2009) as cited in the introduction).

In the following, two components $g_1$ and $g_2$ will be discerned, with $g_1$ the net gain resulting from the fast-feedback processes (the principle determinant for ECS), and $g_2$ referring to the additional slow feedbacks due to the long-term cryosphere response, constituting the basis for the calculation of ESS.

*Equilibrium climate sensitivity ECS*

In their search of an emergent constraint on ECS, Cox et al. (2018) use the simple Hasselmann model (Hasselmann, 1976), relating the variation in global mean surface temperature $\Delta T_s$ in response to a change in radiative forcing $\Delta Q$ by a linear first-order differential equation (see Appendix A for a transfer-function description of this model):

$$C \frac{d\Delta T_s}{dt} = -\lambda \, \Delta T_s + \Delta Q \qquad (4)$$

with C the system's (mainly ocean) heat capacity and $\lambda$ the radiative damping coefficient back to space.

Based on this simple relation, a theoretical emergent relationship for ECS is derived from an ensemble of complex climate models (the CMIP5 archive, see, for instance, Taylor et al. 2012) which can be combined with observed variations of global-mean temperature to produce a constraint on the predicted future change.

The principle parameter of interest to their analysis is the radiative damping coefficient $\lambda$, in their study called the 'climate feedback factor', which with respect to terminology used is somewhat confusing in relation to the feedback factor f and gain





g as defined above by Hansen et al. (2008). In Appendix A a simple reciprocal relation is derived between $\lambda$ and the feedback factor f:

$$f = \lambda_0/\lambda \tag{5}$$

with $\lambda_0$ the radiative damping coefficient in the absence of climate feedbacks (radiative blackbody damping only), which amounts to 4 W m$^{-2}$ K$^{-1}$. With respect to the feedback gain g this yields (right-hand part of Eq. (3)):

$$g = 1 - \lambda/\lambda_0 \quad \text{or, conversely,} \quad \lambda = \lambda_0.(1 - g) \tag{6}$$

In terms of climate sensitivity in the appendix an alternative expression is derived, which by approximation directly relates the fast-feedback factor f (and corresponding feedback gain g) to $\Delta T_{2x}$, the equilibrium mean surface temperature change for a doubling of atmospheric $CO_2$ concentration (being the definition of ECS):

15   $$f \approx \Delta T_{2x} = ECS \qquad \text{and} \qquad g = 1 - 1/f \approx 1 - 1/ECS \tag{7}$$

As an example, for the current IPCC central estimate of ECS = 3 K (from now on referred to as the 'default value') this corresponds to a positive climate feedback gain of $1 - 1/3 = 2/3$. According to the right-hand part of Eq. (6), this reduces the radiative damping coefficient $\lambda$ of the 'Hasselmann system' of Eq. (4) by a factor 3, compared to an idealized blackbody

20   planet.

*Earth system sensitivity ESS*

To obtain an estimate for the 'long-term' Earth system sensitivity, the effect of the slow feedbacks due to the cryosphere
25   response may be incorporated by adding a long-term equilibrium component $g_2$ to the climate fast-feedback gain $g_1$ as described above. This yields the following 'scaling scheme' from ECS to ESS:

| 1. | for a given climate sensitivity ECS the fast-feedback contribution to the gain g is calculated according to Eq. (7) |
|---|---|
| 2. | from this the overall gain $g_c$ is calculated by adding the fast-feedback and long-term equilibrium components: $g_c = g_1 + g_2$ |





| 3. | the combined earth-system feedback factor $f_c$ is derived from $g_c$ according to Eq. (3) |
|---|---|

which results in an estimate of the Earth system sensitivity ESS.

An alternative way to achieve this scaling is based on a direct, nonlinear expression for the combined earth-system feedback

factor $f_c$ (see, for instance, Buchdahl, 1999 and Appendix B for an analytical derivation):

$$f_c = \frac{f_1.f_2}{f_1+f_2 - f_1.f_2} \tag{8}$$

with $f_1$ the net feedback factor of the 'fast' feedbacks and $f_2$ the total feedback factor of the 'slow' second-order feedback processes. In terms of both f and g, in Appendix B it is shown that this combined expression for $f_c$ can be rewritten in a two-stage serial form:

$$f_c = \; f_1 \, . \, \frac{f_2}{f_1+f_2 - f_1.f_2} \qquad = \qquad f_1 \, . \, \frac{1}{1 - f_1.(1 - 1/f_2)} \qquad = \qquad f_1 \, . \, \frac{1}{1 - f_1.g_2} \tag{9}$$

Substituting ECS as an estimate for $f_1$ (first part of Eq. (7)), this may be rewritten as:

$$ESS \approx ECS. \frac{1}{1 - ECS.g_2} \qquad or \qquad ESS/ECS \approx \frac{1}{1 - ECS.g_2} \tag{10a}$$


which results in the estimate of the Earth system sensitivity ESS.

Note that in this equation the scale factor between ECS and ESS is dependent of ECS, expressing the nonlinear nature of this scaling relation.

As a function of the radiative damping coefficient $\lambda$, an alternative formulation is given by (applying Eq.(5)):


$$ESS \approx ECS. \frac{1}{1 - f_1.g_2} \qquad or \qquad ESS/ECS \approx \frac{1}{1 - \lambda_0/\lambda.g_2} \tag{10b}$$

As can be seen from the left-hand part of this expression, the first-stage feedback factor $f_1$ serves as an additional amplification of the second-stage feedback gain $g_2$, illustrating the complex cascading nature of this basic two-stage example.


In the following, the scaling relations as derived here will be applied to investigate the sensitivity of long-term effects to short-term 'deviations' from the ECS default value. Of specific interest are the new constraints put on ECS by Cox et al. (2018) by analyzing the ability of coupled atmosphere-ocean climate models in the CMIP5 archive to simulate observed variations in climate.



## 3 Results

*Equilibrium climate sensitivity ECS*

In Fig. 1 the Equilibrium climate sensitivity ECS is plotted as a function of the radiative damping coefficient $\lambda$, with a maximum value of 4 W m$^{-2}$ K$^{-1}$, the value $\lambda_0$ in the absence of climate feedbacks (radiative blackbody damping only).

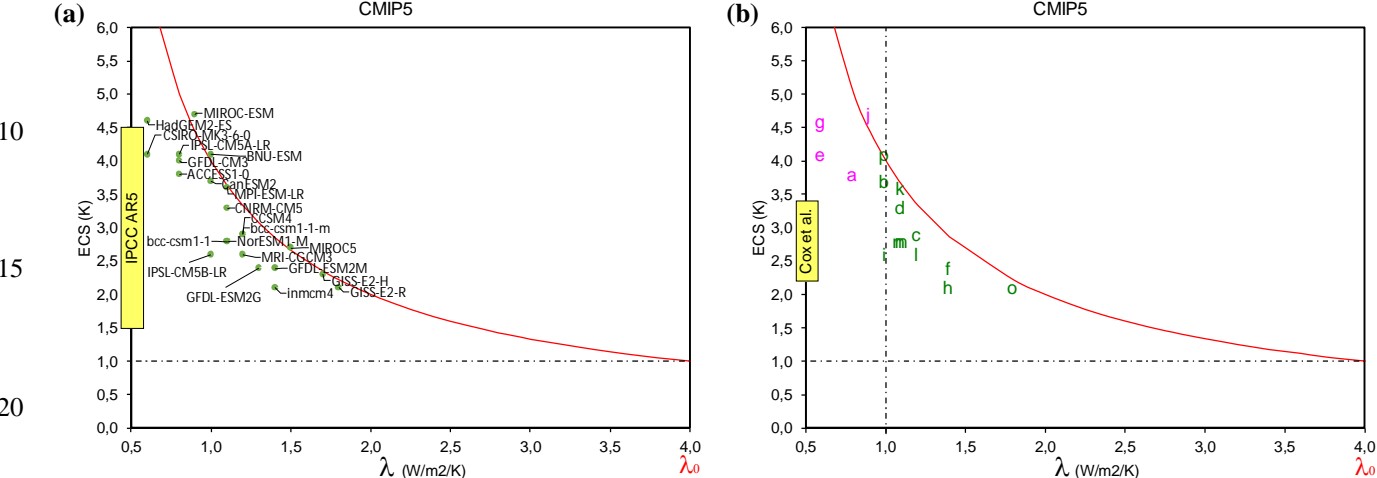

**Figure 1.** ECS as a function of the radiative damping coefficient $\lambda$. The theoretical ECS is plotted against 'raw' model data from the CMIP5 model archive (labeled data points) **(a)** and coded according to Cox et al. (2018) **(b)**. Also, as a vertical yellow bar, the current IPPC 'likely range' of 1.5–4.5 K (left panel) and the 'narrowed down' range of 2.2–3.4 K according to Cox et al. (2018) (right panel) are added.

For each $\lambda$, a 'theoretical' value based on the Hasselmann model for climate sensitivity (red curve) is calculated by the relation

ECS = $\Delta Q_{2x}/\lambda$, with $\Delta Q_{2x} \approx 4$ W m$^{-2}$, the radiative forcing for a doubling of atmospheric $CO_2$ (Appendix A). The horizontal black dot-dashed line (ECS=1) corresponds to the climate sensitivity in the absence of climate feedbacks. To relate this value of ECS to the CMIP5 model archive, in the left panel of the figure the values of $\lambda_i$ and ECS$_i$ for the different CMIP5 models are plotted as labeled green data points, with the labels referring to the model names (see Appendix C). In the right panel the CMIP5 model selection as used by Cox et al. (2018) as input to their analysis are presented, denoted by the letters listed in Table C1. The model points are color-coded as in the Cox study, with lower-sensitivity models ($\lambda > 1$ W m$^{-2}$ K$^{-1}$) shown by green letters and higher-sensitivity models ($\lambda < 1$ W m$^{-2}$ K$^{-1}$) shown by magenta letters. The vertical black dot-dashed line is added to mark the transition for $\lambda=1$. Also, as a reference, the current IPCC ECS 'likely range' (i.e. 66% probability) of 1.5 – 4.5 K (left panel) and the narrowed down range of 2.2 – 3.4 K according to Cox et al. (2018) (right panel) are added as a yellow bar.



*Earth system sensitivity ESS*

In Fig. 2 the estimated long-term Earth system sensitivity ESS is plotted as a function (red curves) of the Equilibrium climate sensitivity ECS for different values of the glacial feedback gain $g_2$, scaled according to Eq.(10a). In the figure this glacial

feedback parameter is denoted as '$g_0$' (in analogy with $\lambda_0$ in Fig. 1). Furthermore, individual scaling results for the CMIP5 models are added as a scatter plot by applying Eq.(10b) to the different values of $\lambda_i$ and $ECS_i$ provided in Appendix C. The model points, denoted by the different letters listed in Table C1, are color-coded as in the Cox study as described above.

As in Fig. 1, the current IPCC ECS 'likely range' (i.e. 66% probability) of $1.5 - 4.5$ K (Fig. 2a) and the narrowed down range of $2.2 - 3.4$ K according to Cox et al. (2018) (Fig. 2b) are added (yellow bars). Using the scaling relation of Eq.(10a) (red

curves), these ECS ranges are mapped onto the corresponding ESS ranges (vertical red bars).

*Choice of glacial conditions:*

With regard to the glacial feedback gain $g_0$, Fig. 2b acts as a reference case ($g_0=0$) for the different glacial conditions depicted in Fig. 2c to 2f, with a maximum value during the 'Last Glacial Maximum (LGM). As described in Hansen (2012), the powerful

glacial albedo feedback of retreating ice-sheets (and corresponding vegetation change) during the transition from this period to the present Holocene was estimated to double the fast-feedback Charney sensitivity of 3 K to an Earth system sensitivity ESS of 6 K. Acccording to Eq.(10a) this requires (substituting ECS=3) a value of $g_0=1/6$ (Fig. 2e and 2f). Furthermore, $CO_2$ forcing was estimated to be amplified by one-third because of non-$CO_2$ GHGs, such as methane release from melting permafrost, acting as a (temperature-dependent) feedback in a warming world (Beerling et al. 2009, 2011). By again applying

Eq.(10a) and substituting ECS=3 K, this would require a gain $g_0=1/12$ (Fig. 2c).

In Hansen et al. (2012) the methane influence is treated as an additional fast feedback, increasing ECS from 3 K to 4 K, and subsequently multiplied by a factor 2 to an ESS of 8 K (blue dot-dashed line in Fig. 2f) to include the LGM glacial albedo feedbacks as described above. However, as also shown in the figure by the red curve, applying the feedback scaling relation of Eq.(10a) for ECS=4 and the LGM slow-feedback gain of 1/6 would result in an ESS of 12 K.

With regard to the slow feedbacks (ice-albedo and vegetation change), in the Hansen paper it is argued that the 'doubling effect' on climate sensitivity during the late Pleistocene epoch from a Holocene perspective is only valid for a negative forcing, because present climate is near the warm extreme of the Pleistocene range. Given the present $CO_2$ concentration of approx. 400 ppm and the corresponding movement toward a warmer climate, a comparison with the mid Pliocene (3 My BP), the last time in Earth history $CO_2$ level was comparable to the present value, might be more relevant. As estimated by Lunt et al. 2010,

climate sensitivity during that period was increased by a factor of 1.3-1.5 by the slow surface-albedo feedbacks, instead of the doubling found for the late Pleistocene. In Fig. 2d a value $g_0=1/9$ was chosen, according to Eq.(10a) for ECS = 3 K corresponding to a feedback factor of 1.5.





**Figure 2.** Earth system sensitivity ESS for the CMIP5 models as a function of ECS. As a first reference, the current IPPC 'likely range' of 1.5-4.5 K is added (yellow bar) **(a)**. Subsequently, the new ECS 'likely range' estimated by Cox et al. (2018) **(b)** serves as a reference for the estimation of ESS (vertical red bar) for different values of the glacial feedback gain $g_0$ **(c)** to **(f)**.



## 4 Discussion and Conclusion

As presented in Fig. 2, the 'narrowing down' of the ECS estimate in the recent Cox et al. study on the longer term may still lead to values in a range (vertical red bars) well beyond the 'original' IPCC range by the 'scaling up' process from ECS to ESS. This is supported by a combined assessment (red curves) of Figs. 1 and 2, showing the rapid (nonlinear) increasing role

of the cryosphere contribution to the overall Earth system sensitivity ESS for a decreasing radiative damping coefficient $\lambda$.

For a given glacial feedback gain $g_0$, according to the left-hand part of Eq.(10b) the scaling relation between ESS and ECS (red curves in Fig. 2) is of the type '$1/(1-f.g_0)$', in close analogy with the fundamental '$1/(1-f)$' behavior of the gain curve as identified by Roe et al. (2007) in their paper on elementary feedback behavior of dynamical systems. As derived in the two-

stage combined feedback analysis of Appendix B, in this expression the (atmosphere-ocean) fast-feedback factor f, besides constituting the basis for ECS, also serves as an additional amplification of the glacial feedback gain $g_0$, possibly affecting the stability of the overall system.

Indeed, from a stability perspective, for '$f.g_0 = 1$' (the loop gain $f_1.g_2$ in Fig. B2 of Appendix B) a singularity arises in the above scaling relation, leading to an 'infinite' value of ESS. Of course, this infinite (or undefined) value of ESS is of a pure theoretical

nature, in terms of Roe indicating that the 'model system has broken down'. In a manner similar to the application of control theory to the linear stability analysis of dynamical systems (see, for instance, Dorf, 1980), a possible useful concept within this context is given by the so-called 'gain-margin' $G_m$:

$$G_m = 1/(f.g_0) \tag{11a}$$


which is the reciprocal of the loop gain described above. This quantity is a (multiplicative) measure of the factor by which the total feedback gain would have to be increased to violate the stability condition of the whole Earth system model. Expressed in terms of the climate sensitivity of the coupled atmosphere-ocean subsystem (DC gain of the Hasselmann model described in Appendix A) a logarithmic (decibel) form may be useful:


$$G_{db} = 20 \log (1/(f.g_0)) = -20 \log (f.g_0) \approx -20 \log (ECS.g_0) \text{ db} \tag{11b}$$

Extending the simple example of Section 2, for the 'default' climate sensitivity ECS of 3 K, with a cryosphere feedback gain $g_0$ of 1/6 (LGM conditions of Fig.2e and 2f) yields: $G_{db} = -20 \log (3/6) = 6$ db, allowing the climate sensitivity of the coupled

atmosphere-ocean system to be doubled from 3 K to 6 K before the stability condition of the overall Earth system model becomes violated (the so-called 'singularity' described above). Treating the methane release from melting premafrost as an additional fast feedback, increasing ECS from 3 K to 4 K according to Hansen et al. (2012), would reduce this quantity from 6 db to 3.5 db, implying a substantial reduction of the stability margin of the overall system. Solely multiplying ECS with a





factor 2 from 4 K to an ESS of 8 K (blue dot-dashed line in Fig. 2f) as was done in the Hansen paper would not account for this effect. In Fig. 2f this is illustrated by the increased curvature of the scaling relation between ESS and ECS, for an ECS of 4 K leading to an ESS of 12 K (red curve).

Also individual scaling results for the CMIP5 models (applying Eq.(10b) to the different values of $\lambda_i$ and $ECS_i$ provided in Appendix C) suggest a relative high sensitivity to an increasing glacial feedback gain $g_0$. In Fig. 2 this is illustrated by the substantially increasing scatter in the different plots from the reference case $g_0 = 0$ (reducing Eq.(10b) to a pure linear scaling relation of "1" between ESS and ECS) to the maximum value of $g_0 = 1/6$, making a distinction between the green letters of the lower-sensitivity models ($\lambda > 1$ W m$^{-2}$ K$^{-1}$) and magenta letters of the higher-sensitivity models ($\lambda < 1$ W m$^{-2}$ K$^{-1}$). Note that

the 'threshold' value of $\lambda \approx 1$ corresponds to an ECS of 4 K, the 'permafrost' value analyzed above. For the LGM condition of $g_0 = 1/6$ (Fig.2e and 2f) some individual models even lead to a 'break down' of the overall model system (in the figure presented as magenta letters with ESS = 0), as described above.

Concluding, using simulation models to determine the required $CO_2$ target to achieve a planetary temperature 'setpoint' like the 2°C or, even more precise, 1.5°C target according to the Paris Agreement, tells us only 'part of the deal': 'unmodeled' cascading (long-term) effects, although relatively small in feedback gain, may push the total Earth climate system towards a new equilibrium well beyond the initial temperature threshold, or, in Hansen's (2012) words, a 'prescription for disaster'.
More in line with the elementary feedback and stability analysis as presented in this work, the precautionary approach to

determine a target value for atmospheric $CO_2$ by Hansen et al. (2008) seems to offer a better alternative: emphasizing the importance of 'long-term' feedbacks on climate equilibrium, and realizing that these second-order feedbacks are already kicking in, a.o. in the form of melting permafrost and (accelerating) ice-sheet disintegration, it was proposed to stabilize Earth's climate by restoring the planet's energy balance as soon as possible on the basis of direct observations, instead of relying too much on simulations (such as in the CMIP5 archive).






**Appendix A: Simple Transfer Analysis**

To simulate and explore the dynamic response of the Earth climate system to variations in radiative forcing, simple models in structure similar to the one proposed by Hasselmann (1976) may be useful. At the core of these models stands a first-order

differential equation, linking changes in radiative forcing $\Delta Q$ to the global mean surface temperature change $\Delta T_s$:

$$C \frac{d\Delta T_s}{dt} = -\lambda \, \Delta T_s + \Delta Q \tag{A1}$$

with C the system's (mainly ocean) heat capacity and $\lambda$ the radiative damping coefficient back to space.

In the Cox et al. (2018) study the Hasselmann model is applied for the stochastic case of white-noise forcing $\Delta Q$, to provide a theoretical basis for their search of an emergent constraint on ECS using observed variations of global-mean temperature.

For the present feedback analysis of climate sensitivity, defined as the equilibrium mean surface temperature change in response to an instantaneous doubling of $CO_2$ concentration, a control-systems approach might be useful to determine the dynamic characteristics of this step response. From this perspective, Eq. (A1) can be considered as the 'planetary thermostat

equation' which after Laplace transformation with the differential operator s yields the following simple transfer function:

$$H_r(s) = \frac{\Delta T_s(s)}{\Delta Q(s)} = \frac{1}{(Cs + \lambda)} \tag{A2}$$

In Fig. A1 the corresponding block diagram is presented as a 'negative-feedback controller' with a feedback gain of $\lambda$, the

radiative damping coefficient. To illustrate the limiting role of the possibly increasing net fast climate feedbacks on the outgoing radiation flux for higher temperatures, the influence on the effective radiative damping coefficient $\lambda$ is expressed as a nonlinear function of $\Delta T_s$. According to the control diagram, the net incoming radiation flux is integrated and stored as additional energy in the system, raising its temperature at a rate determined by the system's heat capacity C. Subsequently, governed by the radiative damping coefficient $\lambda$, the outgoing radiation flux is increased until a new thermal equilibrium is

obtained. An important parameter characterizing this transient response of the first-order system is given by the ratio $C/\lambda$, defined as the time constant or response time $\tau$ (see, for instance, Buchdahl, 1999). Given the large value of the system's (mainly ocean) heat capacity this may lead to a climate response time in the order of magnitude beween 50 and 100 years, as described in Hansen (2005). In equilibrium, as can be directly seen by setting the differential operator s = 0 in Eq.(A2), the (stationary) relation between radiative forcing $\Delta Q$ and global mean surface temperature change $\Delta T_s$ is given by $1/\lambda$, the DC

gain of the above Hasselmann model, which is independent of the system's heat capacity C.



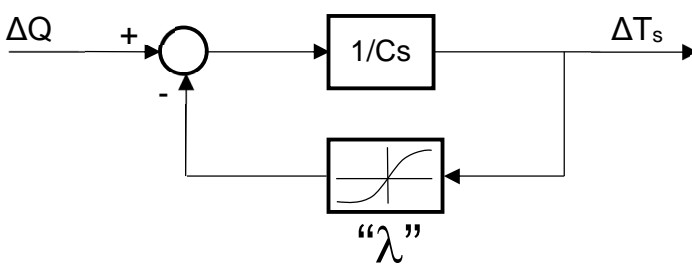

**Figure A1.** Transfer from radiative forcing $\Delta Q$ to global mean surface temperature change $\Delta T_s$ as a 'negative-feedback controller'. According to the diagram, the net incoming radiation flux is integrated and stored as additional energy in the system, raising its temperature at a rate determined by the system's heat capacity C. Subsequently, governed by the radiative damping coefficient $\lambda$, the outgoing radiation flux is increased until a new thermal equilibrium is obtained. In this equilibrium condition, the global mean surface temperature change $\Delta T_s$ is given by $\Delta Q/\lambda$. The nonlinear relation between $\Delta T_s$ and the outgoing radiation flux illustrates the role of fast climate feedbacks in this process.

In Fig. A2, the situation is presented in the absence of climate feedbacks (radiative blackbody damping only). The theoretical value of the radiative damping coefficient for this case, defined as $\lambda_0$, amounts to 4 W m$^{-2}$K$^{-1}$ (see, for instance, NRC (2003) for a derivation, or Roe (2009) for practical implications). In thermal equilibrium, the global mean surface temperature change $\Delta T_0$ is given by $\Delta Q/\lambda_0$. Combining this relation with $\Delta Q/\lambda$ as derived above in the presence of climate feedbacks yields a ratio $\Delta T_s/\Delta T_0$, defined as the feedback factor f, of $\lambda_0/\lambda$.

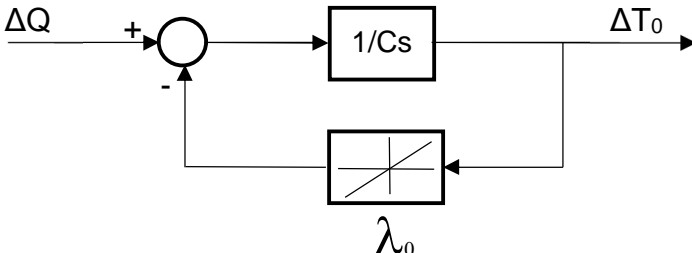

**Figure A2.** Transfer from radiative forcing to global mean surface temperature change in the absence of climate feedbacks (radiative black-body damping only). In thermal equilibrium, the global mean surface temperature change $\Delta T_0$ is given by $\Delta Q/\lambda_0$.

As can be seen from Fig. A1, a special case arises for a forcing $\Delta Q$ of $\lambda_0$ W m$^{-2}$. This forcing more or less equals the radiative forcing of approx. 4 W m$^{-2}$ for a doubling of atmospheric $CO_2$ concentration, defined as $\Delta Q_{2x}$. According to the diagram, for this case the global mean surface temperature change $\Delta T_s$ in thermal equilibrium is given by $\lambda_0/\lambda$, equaling the feedback factor f as defined above. Hence, the equilibrium mean surface temperature change for a doubling of atmospheric $CO_2$ (being the definition of ECS) can be considered as a practical, first estimate of the climate feedback factor f.



## Appendix B: Combined Feedback Analysis

For a given feedback gain g, the corresponding feedback factor f is given by Eq. (3) in Section 2:

$$f = \frac{1}{1-g} \quad \text{or, conversely,} \quad g = \frac{f-1}{f} \tag{B1}$$

For two different gains $g_1$, $g_2$ this yields:

$$g_1 = \frac{f_1-1}{f_1} \quad \text{and} \quad g_2 = \frac{f_2-1}{f_2} \tag{B2}$$

As stated in Section 2, unlike f, a very useful characteristic of the gain g is its additive nature with respect to the individual feedbacks: $g = g_1 + g_2 + \ldots$

Fig. B1 shows the combined amplifying effect of two feedbacks $g_1$ and $g_2$ in the form of an 'equilibrium transfer scheme', which can be considered as the steady state ('DC') equivalent of the dynamic transfer function block diagram, such as presented in Fig. A1.

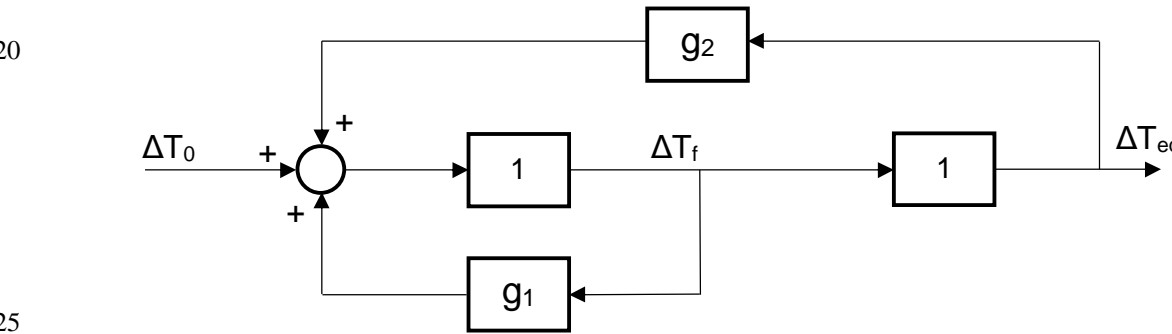

**Figure B1.** Equilibrium transfer scheme for the combined amplifying effects of two feedbacks $g_1$ and $g_2$. The first part of the transfer scheme describes the amplification by the first feedback $g_1$ from $\Delta T_0$, the global mean surface temperature change in the absence of feedbacks, to $\Delta T_f$, the global mean temperature change including the feedback. In the second part of the transfer scheme the amplifying effect of the second feedback $g_2$ is added, resulting in the equilibrium global mean surface temperature change $\Delta T_{eq}$.

In our case of two gains $g_1$ and $g_2$ this yields for the total gain $g_c$:

$$g_c = g_1 + g_2 = \frac{f_1-1}{f_1} + \frac{f_2-1}{f_2} = \frac{f_1.f_2 - f_2 + f_1.f_2 - f_1}{f_1.f_2} = \frac{2.f_1.f_2 - (f_1+f_2)}{f_1.f_2} \tag{B3}$$

and the corresponding combined feedback factor:



$$f_c = \frac{1}{1 - g_c} \tag{B4}$$

Finally, substituting expression (B3) for $g_c$ in Eq. (B4) and rearranging terms yields:

$$f_c = \frac{1}{1 - \dfrac{2.f_1.f_2 - (f_1+f_2)}{f_1.f_2}} = \frac{f_1.f_2}{f_1+f_2 - f_1.f_2} \tag{B5}$$

which is the highly nonlinear, neither additive nor multiplicative relation as reported in Buchdahl (1999).

In terms of both f and g, this combined expression for $f_c$ can be rewritten in the following two-stage serial form:

$$f_c = f_1 . \frac{f_2}{f_1+f_2 - f_1.f_2} = f_1 . \frac{1}{1 - f_1.(1 - 1/f_2)} = f_1 . \frac{1}{1 - f_1.g_2} \tag{B6}$$

As can be seen from the right-hand part of this expression, the first-stage feedback factor $f_1$ serves as an additional amplification of the second-stage feedback gain $g_2$, showing the complex cascading nature of this basic two-stage example as illustrated below in Fig. B2.

**(a)**

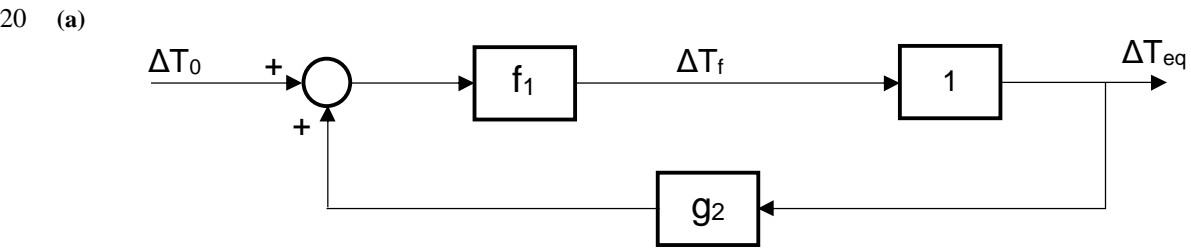

**(b)**

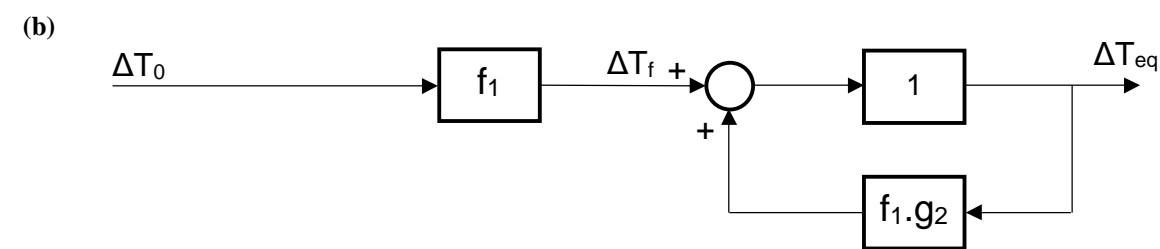

**Figure B2.** Combined representation of the equilibrium transfer scheme in terms of both the feedback factor f and feedback gain g. As a first step **(a)** the feedback gain $g_1$ of Fig. B1 is replaced by the corresponding feedback factor $f_1$. Subsequently, this is rearranged in the equivalent two-stage serial form from input to output **(b)**.



## Appendix C: CMIP5 Models

**Table C1.** Radiative damping coefficients $\lambda$ and Equilibrium Climate Sensitivity ECS for the different Earth system models used as input to the Cox et al. (2018) study, as provided by the CMIP5 archive.

| | Model | $\lambda$ (W m$^{-2}$ K$^{-1}$) | ECS (K) |
|---|---|---|---|
| **a** | ACCESS1-0 | 0.8 | 3.8 |
| **b** | CanESM2 | 1.0 | 3.7 |
| **c** | CCSM4 | 1.2 | 2.9 |
| **d** | CNRM-CM5 | 1.1 | 3.3 |
| **e** | CSIRO-MK3-6-0 | 0.6 | 4.1 |
| **f** | GFDL-ESM2M | 1.4 | 2.4 |
| **g** | HadGEM2-ES | 0.6 | 4.6 |
| **h** | inmcm4 | 1.4 | 2.1 |
| **i** | IPSL-CM5B-LR | 1.0 | 2.6 |
| **j** | MIROC-ESM | 0.9 | 4.7 |
| **k** | MPI-ESM-LR | 1.1 | 3.6 |
| **l** | MRI-CGCM3 | 1.2 | 2.6 |
| **m** | NorESM1-M | 1.1 | 2.8 |
| **n** | bcc-csm1-1 | 1.1 | 2.8 |
| **o** | GISS-E2-R | 1.8 | 2.1 |
| **p** | BNU-ESM | 1.0 | 4.1 |
| **f[x]** | GFDL-ESM2G | 1.3 | 2.4 |
| **f[y]** | GFDL-CM3 | 0.8 | 4.0 |
| **i[x]** | IPSL-CM5A-LR | 0.8 | 4.1 |
| **j[x]** | MIROC5 | 1.5 | 2.7 |
| **n[x]** | bcc-csm1-1-m | 1.2 | 2.9 |
| **o[x]** | GISS-E2-H | 1.7 | 2.3 |



## Appendix D: List of variables and parameters

| Symbol | Description | Units |
|--------|-------------|-------|
| ECS | Equilibrium climate sensitivity | K |
| ESS | Earth system sensitivity | K |
| S | Climate sensitivity | $K\ W^{-1}\ m^2$ |
| $S_{CO2}$ | Earth system sensitivity to $CO_2$ forcing | $K\ W^{-1}\ m^2$ |
| F | Specified unit radiative forcing | $W\ m^{-2}$ |
| $\Delta Q$ | Radiative forcing | $W\ m^{-2}$ |
| $\Delta Q_{2x}$ | Radiative forcing for a doubling of atmospheric $CO_2$ concentration | $W\ m^{-2}$ |
| $\Delta T_{2x}$ | Equilibrium mean surface temperature change for a doubling of $CO_2$ | K |
| $\Delta T_{eq}$ | Equilibrium global mean surface temperature change | K |
| $\Delta T_0$ | Global mean surface temperature change in the absence of climate feedbacks | K |
| f | Climate feedback factor | unitless |
| g | Climate feedback gain | unitless |
| $g_0$ | Glacial feedback gain in the absence of climate feedbacks | unitless |
| $g_c$ | Combined gain for multiple feedbacks | unitless |
| $\lambda$ | Radiative damping coefficient | $W\ m^{-2}\ K^{-1}$ |
| $\lambda_0$ | Radiative damping coefficient in the absence of climate feedbacks | $W\ m^{-2}\ K^{-1}$ |
| s | Laplace variable (differential operator) | |
| $H_r(s)$ | Radiation transfer function | |
| $\Delta T_s$ | Global mean surface temperature change | K |
| C | Earth system's heat capacity | $J\ K^{-1}\ m^{-2}$ |
| $\tau$ | Time constant of the Earth system's transient climate response | s |
| $\Delta T_f$ | Global mean surface temperature change including fast (hydrological) feedbacks | K |
| $f_c$ | Combined feedback factor for multiple feedback gains | unitless |
| $G_m$ | Gain margin of the total Earth climate system | unitless |
| $G_{db}$ | Logarithmic (decibel) expression of the Gain margin | db |





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
