# Peer review of "Earth System Sensitivity: a Feedback perspective"

_Earth System Dynamics, 2021_

## Author Comment (AC1)

**Author Response to RC1**

Thank you very much for your very useful and to-the-point comments to my manuscript. Although you conclude that the paper presents some interesting ideas (from the world of electronics) you find it difficult to follow and you think it needs to account for more of the recent progress concerning emergent constraints (in the field of climate research).
Below, my responses and corresponding (proposed) changes to the manuscript are provided, with your original comments in italics.

**Comment**

*In the paper Earth System Sensitivity: a Feedback perspective, Peter O. Passenier discusses emergent constraints for equilibrium climate sensitivity (ECS) and argues that slow feedbacks (e.g. permafrost and ice sheet dynamics) are not properly accounted for in previous work. The paper is short - which is nice in many respects, but I also have a concern whether it adds new information. Science papers need to explain the current state of science on the chosen topic to demonstrate that they are up to date (scholar.google search with '"emergent constraints" AND ECS' gave 239 hits, many of which were published since 2018 - most of the cited literature herein are older than those). This manuscript doesn't do that. It may nevertheless, present some new ideas and insight, but I'm not able to say if it is or isn't. Analogies from the world of electronics, however, are interesting and probably quite novel within climate research.*

**Response**

An important aim of my study is to support a better mechanistic understanding of interactions *in* (as opposed to improving the predictability *of*) the Earth climate system (see the Introduction). The starting point was the Cox et al. study (2018), who used the simple but elegant Hasselmann (1976) model as a basis for their emergent constraints 'update' of the ECS 1.5-4.5 K likely range, the well-familiar IPCC AR5 (2013) range already dating back to Charney (1979). I think, given the mainly pedagogic purpose of my paper, the 2018 references made still can be justified, although as you rightly point out there have been recent important community developments in this area which at least should be mentioned in the Introduction of my study (an overview of more recent research is provided in Sherwood et al. 2020 and IPCC AR6 WG1, see RC2)

**Proposed changes**

The Introduction section shall be extended, placing the Cox et al. 2018 and IPCC AR5 references in the more recent (Sherwood et al. 2020 and IPCC AR6) context as described above.

**Comment**

*Another question is whether some of the derivations and mathematics presented in the Methods section should be left in an appendix.*

**Response**

The idea behind the division between the Method section and the Appendices is to separate the 'standard' feedback analysis parts ('Simple transfer analysis' and 'Combined feedback analysis') from the main body of the paper, where these techniques are applied as a method to investigate the interactions between climate sensitivity ECS and earth system sensitivity ESS. My approach was to use the 'stochastic' Hasselmann (filter) model in a 'deterministic' (control) mode (Appendix A of the paper) as a methodological basis for the combined feedback analysis, described in more detail in Appendix B.

**Comment**

*It is possible that slow feedbacks also affect the fast ones and that the dynamics and thermodynamics involve nonlinear interactions so that the total feedback no longer is the sum of individual feedbacks. Hence, the paper assumes that the effect from various processes are additive, which I don't think has been convincingly demonstrated. The paper does, however, discuss combined earth-system feedbacks in the context of earth system sensitivity. I think that this part needs to be explained more carefully.*

**Response**

I agree with you that in 'reality' the interactions mentioned are probably highly nonlinear. A common approach within the community research on climate feedbacks and sensitivity (see for instance IPCC AR6 WG1 Chapter 7) is to decompose, to first order, the net feedback parameter (in my paper defined as the 'radiative damping coefficient') into a sum of terms. This also constituted the methodological basis for my extension to the assessment of earth system sensitivity. As demonstrated in the paper however, already in this simple case of additive feedback gains $g_i$ the combined scaling relation between input and output (feedback factor $f_c$) becomes highly nonlinear, with possible consequences for system stability.

**Comment**

*I find it a bit hard to see the 'red thread' in this paper, which presents a selection of 'facts' without sufficient context or explanation for why. It would be easier to follow the train of thoughts with a clearly stated hypothesis and explicit definitions. Explain why the mathematical derivations and why presenting e.g. Fig 1. It doesn't suffice doing so only in the introduction.*

**Response**

Realizing that a lot of the above considerations, though (implicitly) addressed in the manuscript, are more or less hidden to the reader, I propose an 'Outline' section between the (extended, see my first response above) Introduction and the Method section. This section should serve as a more explicit guide in defining the main research questions and managing reader expectations to answer these questions throughout the rest of the paper.

**Proposed changes**

Adding of an 'Outline' section between the extended Introduction and the Method section.

**Comment**

*In conclusion, the paper presents some interesting ideas, but I find it difficult to follow and think it needs to account for more of the recent progress concerning emergent constraints. Also, a more careful guidance through the ideas and concepts will make the paper easier to follow. It is always a bit more difficult to follow interdisciplinary work because some aspects often are a bit unfamiliar. Here, the paper relied on ideas from electronics in addition to maths.*

**Response**

Indeed, the paper has been written in a way which implicitly assumes some familiarity with the basic concepts of feedback analysis, originating from electrical (control) engineering, as described by Roe in his pedagogic review of 2009 (see the Introduction of my paper and reference below).

I hope my responses and proposed changes above are sufficient and adequate to solve these issues mentioned by you.

**Comment**
*Minor*; 'IPPC' should be 'IPCC'.

**Response**
Noted

**References**

IPCC AR6 WG1 Chapter 7:
https://www.ipcc.ch/report/ar6/wg1/downloads/report/IPCC_AR6_WGI_Chapter_07.pdf

Sherwood et al. (2020). An assessment of Earth's climate sensitivity using multiple lines of evidence. Reviews of Geophysics, 58, e2019RG000678.
https://doi.org/10.1029/2019RG000678

Roe, G.H. (2009). Feedbacks, Timescales and Seeing Red. Annu. Rev. Earth Planet Scie, Vol. 37:93-115. http://dx.doi.org/10.1146/annurev.earth.061008.134734

---

## Author Comment (AC2)

**Author Response to RC2**

Thank you very much for your very valuable and constructive review of my manuscript. Although you find the manuscript well written and interesting to read, you have several major concerns that need to be addressed before you could recommend publication in ESD. Below, my responses and corresponding (proposed) changes to the manuscript are provided, with your original comments in italics.

**Comment**
*1) Sherwood et al. 2020 and IPCC AR6 WG1 Chapter 7 are two recent community assessments of ECS that narrow the likely range to about 2.5-4 K based on multiple lines of evidence (including emergent constraints). This is the likely ECS range that should be used in the analysis instead of the single study of Cox et al. 2018 (which has been challenged on methodological grounds and may turn out to not be robust, see discussion in IPCC AR6 Chapter 7).*

**Response**
The starting point for my study was the Cox et al. study (2018), who used the simple but elegant Hasselmann (1976) model as a basis for their emergent constraints 'update' of the ECS 1.5-4.5 K likely range, the well-familiar IPCC AR5 (2013) range already dating back to Charney (1979). Indeed, as rightly mentioned in your comment above (with the references you provided), since 2018 important community assessments of 'emergent constraints' have taken place, which should of course be mentioned in the Introduction of my study.

An important aim of my study is to support a better mechanistic understanding of interactions *in* (as opposed to improving the predictability *of*) the Earth climate system (see the Introduction). I think, given this mainly pedagogic purpose, my 2018 references on ECS ranges in relation to the Hasselmann model still can be justified as a basis for the rest of my paper.

**Proposed changes**
The Introduction section shall be extended, placing the 'outdated' Cox et al. 2018 and IPCC AR5 references in the more recent (Sherwood et al. 2020 and IPCC AR6) context as described above.

**Comment**
*2) The AR6 definition of ECS includes everything but the feedbacks associated with ice sheet changes and CO2. That is, it includes methane, vegetation, and many other biogeochemical/physical feedbacks whose values are assessed in AR6 Chapter 7. So, the only feedback of relevance for ESS here would be the ice sheet feedback.*
*Regarding the ice sheet feedback, AR6 Chapter 7 states the following:*
*… ice sheet mass loss leads to fresh water fluxes that can modify ocean circulation (Swingedouw et al., 2008; Goelzer et al., 2011; Bronselaer et al., 2018; Golledge et al., 2019). This leads to reduced surface warming… However, model simulations in which the Antarctic ice sheet is removed completely in a paleoclimate context indicate a positive global mean feedback on multi-millennial timescales due primarily to the surface albedo change… This net positive feedback due to ice sheets on long timescales is also supported by model simulations of the mid-Pliocene warm period… As such, overall, on multicentennial timescales the feedback parameter associated with ice sheets is likely negative (medium confidence), but on multi-millennial timescales by the time the ice sheets reach equilibrium, the feedback parameter is very likely positive (high confidence; see Table 7.10). However, a relative lack of models carrying out simulations with and without interactive ice sheets over centennial to millennial timescales means that there is currently not enough evidence to quantify the magnitude of these feedbacks, or the timescales on*

*which they act.*
*That is, on timescales of a century (of relevance for the Paris targets) the ice sheet feedback is probably negative, and only on timescales of several centuries and longer does it become positive, but with a value that is not well quantified on either timescale. In light of the AR6 assessment, a positive ice sheet feedback does not seem to be relevant for Paris targets. The ice sheet feedback values chosen in this study need to be well justified. And note that the value derived from the LGM is not suitable for a calculation of ESS relevant for future warming.*

**Response**
I agree with your comment that the positive ice-sheet feedback in itself does not seem to be relevant for Paris targets (to limit the warming in the pipeline for this century to some preset value), given the different timescales as described in your AR6 quote of section 7.4.2.6. On the other hand, as stated in the first part of this AR6 section:
*….Although long-term radiative feedbacks associated with ice sheets are not included in our definition of ECS 52 (Box 7.1), the relevant feedback parameter is assessed here because the timescales on which these feedbacks act are relatively uncertain, and the long-term temperature response to CO2 forcing of the entire Earth system may be of interest….*
According to my study, Paris targets (and ECS) may have a large influence on the potential effect of this ice-sheet feedback parameter on the (committed) long-term temperature response (ESS), as the CMIP5 simulation results for the different ECS ranges and ice-sheet feedback values suggest (Fig.2 of the manuscript).
The LGM value was chosen to be able to relate these results to the Hansen (2012) study on *Paleoclimate Implications for Human-Made Climate Change* (referred to in the manuscript), deriving estimates of different values of ESS based on changes in LGM boundary forcings. The essence of this part of my analysis is showing what would happen if these changes are treated as combined feedbacks.
Please also see my response to your, in my eyes highly related, next comment, asking for a better explanation of the relevance and novelty of the feedback analysis performed in the study.

**Proposed changes**
A better explanation of the above rationale behind the choice of the different glacial feedback values as described in Section 3 of the manuscript.

**Comment**
*3) Given the above, I am not sure that the analysis can add much to the existing literature on ESS. While showing the impact of different hypothetical ice sheet feedback values for ESS (on top of different ECS ranges) would be a fine exercise, I don't see what new information it would provide or how ESS relates to Paris targets which deal with warming this century. It's also possible that I am missing something, but either way the author needs to better explain the relevance and novelty of the feedback calculation performed here.*

**Response**
As stated in the Introduction of my manuscript the principal aim of the study is pedagogic, in exploring possible consequences of the narrowing down of ECS estimates for the long-term Earth system sensitivity (ESS), taking into account 'slow' feedbacks due to the cryosphere response to a warming world. Possible consequences for/of Paris targets are discussed, given the current community focus on the 'allowable' amount of warming this century (which to me seems a rather arbitrary choice) using model-based integrated impact assessments as a basis for international policy making. This, while signs are already present of a destabilization of the cryosphere kicking in (either leading to a temporary negative but in the end positive feedback gain), virtually certain causing large changes beyond the 'Paris time

horizon', on an uncertain timescale. Thus, having said this, the relation of ESS to Paris targets becomes at least twofold:

- On a century timescale (the 'Paris horizon') cryosphere feedbacks may already kick in, one of them being the natural release of non-CO2 trace gases (such as methane, in my study referred to as 'permafrost').

- Beyond the century (millennium?) timescale the ice-sheet albedo feedback comes into play. Current century Paris targets and ECS ranges are more or less setting the stage for the magnitude of the long-term temperature response (ESS) caused by this feedback, as described in my above response to your comment 2*)*

As mentioned in my study, Hansen (2012) introduced SCO2, the 'Earth system sensitivity to CO2 change', with both natural changes of non-CO2 trace gases and ice-sheet albedo effects counted as feedbacks. His approach was to first isolate the role of the non-CO2 trace gases, by treating them as 'fast feedbacks' (thus increasing ECS, relevant for the 'Paris horizon' as mentioned above in the first part of my comment) and subsequently incorporate the effect of the ice-sheet feedback, relevant for ESS (second part of my comment). The focus for this exercise was the LGM, with the note that the estimate would be smaller for positive forcings with the Holocene as the initial state.

As results of my study show (Fig.2e and f of the manuscript), compared to this 'sequential feedback scheme', especially for higher ECS values the combined feedback approach leads to quite different results for ESS, more in line with the fundamental work of Roe (2007, 2009) on the predictability of climate sensitivity.

**Proposed changes**
Realizing that a lot of the above considerations, though (implicitly) addressed in the manuscript, are more or less hidden to the reader, I propose an 'Outline' section between the (extended, see response 1) Introduction and the Method section. This section should serve as an explicit guide in managing reader expectations throughout rest of the paper (also see my response to RC1).

**References**

IPCC AR6 WG1 Chapter 7:
https://www.ipcc.ch/report/ar6/wg1/downloads/report/IPCC_AR6_WGI_Chapter_07.pdf

Sherwood et al. (2020). An assessment of Earth's climate sensitivity using multiple lines of evidence. Reviews of Geophysics, 58, e2019RG000678.
https://doi.org/10.1029/2019RG000678

Hansen, J. and Sato, M. (2012). Paleoclimate Implications for Human-Made Climate Change. Climate Change, Inferences from Paleoclimate and Regional Aspects, A. Berger et al. (eds.), Springer-Verlag Wien, 21-47. http://dx.doi.org/10.1007/978-3-7091-0973-1_2